# Procurement of Uterus in a Deceased Donor Multi-Organ Donation National Program in France: A Scarce Resource for Uterus Transplantation?

**DOI:** 10.3390/jcm11030730

**Published:** 2022-01-29

**Authors:** Ludivine Dion, Gaëlle Santin, Krystel Nyangoh Timoh, Karim Boudjema, Louise Jacquot Thierry, Tristan Gauthier, Marie Carbonnel, Jean Marc Ayoubi, François Kerbaul, Vincent Lavoue

**Affiliations:** 1Department of Gynecology, Hôpital Sud, Rennes University Hospital, CEDEX 9, 35200 Rennes, France; krystel.nyangoh.timoh@chu-rennes.fr (K.N.T.); louise.jacquot.thierry@chu-rennes.fr (L.J.T.); vincent.lavoue@chu-rennes.fr (V.L.); 2UMR_S 1085, IRSET-INSERM, 35000 Rennes, France; 3Organ and Tissue Transplant Management, Biomedicine Agency, CEDEX, 93212 La Plaine-Saint-Denis, France; gaelle.santin@biomedecine.fr (G.S.); francois.kerbaul@biomedecine.fr (F.K.); 4Department of Hepatobiliary Surgery and Liver Transplantation, Pontchaillou, Rennes University Hospital, CEDEX 9, 35000 Rennes, France; karim.boudjema@chu-rennes.fr; 5Department of Gynecology, Limoges University Hospital, 87042 Limoges, France; tristan.gauthier@chu-limoges.fr; 6Department of Obstetrics Gynecology and Reproductive Medicine, Foch Hospital, 92150 Suresnes, France; m.carbonnel@hopital-foch.com (M.C.); jm.ayoubi@hopital-foch.org (J.M.A.)

**Keywords:** uterus transplantation, absolute uterine infertility, ideal donor, expanded criteria donor

## Abstract

Uterus transplantation is a new possibility for women suffering from absolute uterine infertility to become pregnant and have children. In the case of a deceased donor, a list of exclusion criteria is defined to ensure the high quality of the uterus graft. This study evaluates the number of potentially available uterus grafts based on the pre-defined exclusion criteria in a national deceased donor multi-organ donation program in France. We analyzed the data reported in the CRISTAL database regarding all women aged 18 to 60 on whom organ procurement was performed between 2014 and 2019. Potential deceased women donors were classified into three categories: very ideal donor, ideal donor, and expanded criteria donor. Between 2014 and 2019, 4544 women underwent organ procurement. Using the very ideal donor, ideal donor, and expanded criteria donor classification, we found that, respectively, only 124, 264, and 936 donors were potentially eligible for UTx. This represents 2.8 per million people (PMP) very ideal donors, 3.8 PMP ideal donors, 8.6 PMP expanded criteria donors (ECDs). The restricted number of grafts requires a complementary strategy of living and deceased donors to meet the demand of all women with AUI.

## 1. Background

Uterus transplantation (UTx) is a new possibility for women suffering from absolute uterine infertility (AUI) to have children. For these women, it is an alternative to surrogacy or adoption. Therefore, UTx is the only possibility for them to be gestational, genetic, and legal mothers. Since the first livebirth after UTx was obtained by the Brännström team in 2014 [1], more than 70 UTx have been performed, and 27 children were born [2]. Two potential origins of the graft have been developed by several teams: living donors [3,4,5,6,7], deceased donors (DD) [8,9,10], or both [11,12]. Using a living donor is the option for which we have the most experience [3], but in 2019, the first birth after UTx from a deceased donor occurred in Brazil [10], followed by others in the USA [13,14] and Czech Republic [15]. Mayer–Rokitansky–Küster–Hauser (MRKH) syndrome represents the most frequent indication for UTx, estimated at more than 85% of all UTx [12]. In the general population, the prevalence of MRKH syndrome is around 1 in 4500 women or 220 per million people (PMP). In France, the MRKH women aged between 18 and 38 years could represent around 1860 women. In a recent survey conducted among French patients who suffer from AUI, 58% of the respondents were interested in UTx, despite the complexity of the pathway, compared with surrogacy or adoption [16]. Other indications for UTx are less represented, such as hysterectomy for post-partum hemorrhage, estimated at 1 per 1000 births [17], hysterectomy for cervical cancer, or a non-functional uterus, as in Asherman’s syndrome. Thus, we can estimate that more than 1000 women could be candidates for UTx in a country such as France, whose population is more than 65 million people.

Del Priore et al. first described the process of uterus retrieval from a multi-organ donor in 2007 in the state of New York [18], concluding that “transplantation medicine and surgery have advanced to allow nonvital organ allografts”. A second team has shown the feasibility of the surgery, especially the modification of the flushing system by using bilateral femoral artery catheters [19]. Operating time is short: 10 to 30 min [18,19]. Using a deceased donor has the main advantage of not causing injury to the donor. Furthermore, many women with AUI might not have a living donor, for a variety of reasons (strict immunologic or medical exclusion criteria of the donor, donor afraid of the surgery, etc.). The advantages and disadvantages of both types of donors are summarized in Table 1. Deceased donors are a source of potential uterine grafts, but few data are available regarding the potential number of uterine grafts available in multi-organ donation programs and thus the number of uterus grafts available for UTx. In the case of a deceased donor, a list of exclusion criteria is defined to ensure the high quality of the graft, but these criteria decrease the number of potentially available uterus grafts. In other words, a higher number of criteria ensures higher quality grafts but reduces their availability.

The aim of this study is to evaluate the number of potentially available uterus grafts based on the pre-defined exclusion criteria, in the context of a national deceased donor multi-organ donation program in France.

## 2. Material and Methods

### 2.1. Survey Development

In France, the data of all people that undergo a multi-organ donation process are collected by the Biomedicine Agency. A regional agency collects local data, which are then listed nationally in a register named CRISTAL. After agreement from the French National Agency of Biomedicine, we queried the CRISTAL listing and collected the data of women who underwent multi-organ procurement. We analyzed the data reported in CRISTAL regarding all women aged 18 to 60 years on whom organ procurement was performed between 1 January 2014 and 31 December 2019. This study is exempt from approval by an ethics board and was approved on 9 March 2020 by the French National Agency of Biomedicine.

### 2.2. Inclusion Criteria

Potential deceased female donors were classified into three categories: very ideal donor, ideal donor, and expanded criteria donor, as described in the literature [20], and adapted to the data available in the French database. These criteria are detailed in Table 2. Compared with the study of Kristek et al. [20], we arbitrarily added a patient category: the very ideal donor. This addition was made for the sole purpose of knowing the number of grafts available by having very strict donor inclusion criteria, with a BMI < 25 and the absence of antecedent cesarean section. Regarding tobacco use, the cutoff was set at the threshold for which there is a significantly greater risk of stroke, i.e., 20 pack years [21]. In the CRISTAL database, tobacco consumption is given in pack years and not in cigarettes/day. To adapt, the authors sought a cutoff of tobacco consumption that could lead to higher risks of cardiovascular events.

Lubin et al. [21] described a greater increase in cardiovascular pathology starting from 20 PY, which is why this cutoff was defined (RR 1.62 95%CI 1.4–1.8).

### 2.3. Exclusion Criteria

Exclusion criteria were defined based on those found in the literature [20] and are summarized in Table 2.

### 2.4. Design of the Study

We calculated the number of available uterus grafts for each kind of donor (ideal or expanded criteria) three ways: over the whole period, per year, and per million people per year. According to INSEE, the French population is more than 67 million people, of which 34 million are women, with more than 17 million women aged 18 to 60 years.

We calculated the number of potential deceased donors per million people of the total population. This result was compared with the number of women who suffer from AUI per million people

## 3. Results

Between 2014 and 2019, 4544 women underwent organ procurement. The maximal number of potential grafts after exclusion of women with a body mass index (BMI) over 35 kg/m^2^, women with positive serologies (HIV, HCV, and HBV), and women with cardiovascular diseases by age category are represented in Figure 1. 

Very ideal donors were determined as follows: 326 women aged 18 to 35 years, among which 240 had a BMI lower than 25 kg/m^2^, of which 238 had no active infection from HIV, HCV, HBV, HSV, or syphilis. Only 128 of them had no history of smoking, further reduced to 124 women with no cardiovascular disease. Over the study period, only 124 very ideal donors were identified.

Ideal donors were determined as follows: 668 women aged 18 to 45 years, among which 572 had a BMI lower than 30 kg/m^2^, of which 568 had no active infection from HIV, HCV, HBV, HSV, or syphilis. Only 286 of them had no history of smoking, further reduced to 264 women with no cardiovascular disease or only medical treatment. Over the study period, only 264 ideal donors were identified.

Expanded criteria donors were determined as follows: 1830 women aged 18 to 60 years, among which 1694 women had a BMI lower than 35 kg/m^2^, 1666 of which had no active infection from HIV, HCV, HBV, HSV, or syphilis. Only 1109 of them had a smoking history of less than 20 pack years, further reduced to 936 women with no cardiovascular disease or only medical treatment. Over the study period, only 936 expanded criteria donors were identified.

Using the very ideal donor criteria, on average, per year, only 124 donors were potentially eligible for UTx, i.e., 21 women by year or 2.8 women PMP. By expanding criteria inclusion especially regarding age, BMI, and active smoking status, 936 donors were potentially eligible for UTx, i.e., 156 by year or 8.6 PMP. The numbers of potential very ideal, ideal, and expanded criteria donors by year between 2014 and 2019 are summarized in Table 3.

## 4. Discussion

We showed that 4544 deceased donors could be eligible for uterus donation, at most, in terms of very expanded criteria donor, it represents 8.6 PMP. This is very insufficient, compared with potential demand, estimated at 220 PMP.

In France, compared with other organs, the number of uterine grafts eligible for transplantation from deceased donors is low. In 2019, the number of uterine transplants from an ideal donor was estimated at 3.8 PMP; for the same year, it was 55.3 PMP for kidney, 21 PMP for liver, 6.6 PMP for heart, 5.8 PMP for lung, and 1.17 PMP for pancreas [22]. Compared with other organs, the number of PMPs for planned uterus donors is higher than for pancreas donors. In pancreas transplantation, vascular thrombosis is the most common nonimmunologic cause of early pancreas transplant failure, with an estimated loss rate of approximately 5–29%, and usually occurs in the splenic vein [23]. Similar results are observed in the UTx [12]. However, in the near future, the graft loss rate can be expected to become much better than that of the pancreas, for example, by improving organ preservation or expanding the criteria for deceased donors.

In the United States, the number of people on the transplant waiting list by organ type is available on the online database Organ Procurement and Transplantation Network. According to this database, the number of patients waiting for uterus transplants in the United States is seven to date; this number is low, compared with the number of patients waiting for heart–lung transplants, which is 39. The number of patients waiting for kidney transplants is much higher, with 97,000 [24]. The number of people waiting for kidney transplants over the last 10 years is increasing, while the number of people waiting for liver transplants is rather stable [25].

Due to the recent development of the UTx program, no inclusion criteria were clearly defined in the literature. For example, regarding the age of the donor, a Dallas team of researchers collected data from women under 35, while the maximum age for the Czech Republic team was 60. Three proposals for deceased donor inclusion criteria are summarized in Table 4.

Using the strict criteria given by the Dallas team, only 124 women could have donated their uterus during a period of 5 years, i.e., only up to 2.8 PMP. A recent study conducted in the Czech Republic, Sweden, the United Kingdom, and the United States (between 2012 to 2016) estimated the number of potential deceased donors at 2.33 PMP, for a recipient prevalence of 250 PMP [20]. This result is comparable to our findings; similar to our study, the authors of this previous study highlighted the apparent lack of potential deceased donor (DD) grafts for all recipients. To increase the number of available uteri from deceased donors, the inclusion criteria for donors should be expanded.

Regarding UTx, we observed that the number of donors is lower than the potential applications. In our study, broadening the criteria for donor inclusion increases the number of potential transplants by a factor of seven (936 for expanded criteria donors vs. 124 for very ideal donors). The main advantage of using a very ideal donor graft is to obtain a better quality graft. Broadening the inclusion criteria could lead to lower quality grafts and, therefore, lower chances of successful grafting and, thus, pregnancy rate or childbirth.

For example, while in the Cleveland team’s study, the authors specified an upper limit of 30 kg/m^2^ BMI as one of their exclusion criteria for DD [26], in the first published case of live birth from a deceased donor’s uterus in the United States, the BMI of the donor was 31 kg/m^2^ [13]. Similarly, the Dallas team published their criteria for deceased donor uterus donation and specified exclusion criteria, such as an upper age limit of 35 years [27]; however, the two UTx procedures performed by this team used, in fact, donors older than 35 (39 and 48 years old) [12]. This shows that the lack of grafts makes the extension of donor criteria necessary. However, should the donor criteria be expanded, lower quality grafts are to be expected. For instance, a German team failed to perform a transplant due to the extreme resistance of a uterine artery [6]. In this particular case, the team used a living donor, who was 61 years old, postmenopausal, an active smoker (20 pack years), and suffered from metabolic syndrome (overweight with a BMI of 30.9 kg/m^2^, a pharmacologically controlled arterial hypertension with ACE inhibitor, diuretic and imidazoline derivative, and type 2 diabetes under dietary control). Even if the donor had a thorough uterine check-up with a magnetic resonance angiography (MRA) and a contrast-enhanced computed tomography (CT), the graft was not compatible with the transplant because of the extremely high risk of graft vessel thrombosis. In the case of a deceased donor, the check-up is less thorough, and therefore, the risk of poor-quality grafts or loss of the graft is higher. The lesson to be learned from this case is that a combination of cardiovascular diseases is a contraindication for transplantation even if the uterus check-up is normal.

In 2017, the Czech Republic team published nine cases of uterus transplantation—five from living donors and four from deceased donors [11]. Among the six viable uteri, only one child was born [15]. Despite several embryo transfers [28], several early stage miscarriages were observed. This could be explained by the donor criteria, as, in this study, donors were eligible up to the age of 60 [11], whereas previous teams used younger donors: premenopausal for the Cleveland team [26] and younger than 35 years for deceased donors in the Dallas team [27]. No inclusion or exclusion criteria are standardized for UTx with DD; thus, the international UTx registry will help to standardize practices and determine the best criteria for achieving chances of successful transplantation (return of menstruation) and pregnancy. Used criteria to define very ideal, ideal, and expanded deceased donors in the present study were based on a review of the literature and primary experiences in UTx all over the world. These criteria could be changed in the future with rising experience in the UTx field and the international UTx registry in process.

In our study, the gynecological history of each patient (i.e., parity, modalities of delivery) was not detailed in the CRISTAL database. Nevertheless, these data seem to be important to judge the ability of the uterus to develop a pregnancy, even if the Czech Republic team published the first live birth after a nulliparous DD [15]. In France, these data can be recovered from the donor’s family. Organ procurement is possible if the donor has expressed the wish to donate their organs during their lifetime or did not express refusal during their lifetime, and after agreement from the family.

The restricted number of potential deceased donors is a limitation of this type of donor. To overcome this limitation, some teams use both living and deceased donors, such as the Dallas [12], Czech Republic [11], or Sweden teams [29]. The two approaches are complementary because each type of donor has advantages and disadvantages. For example, when a uterus is donated from a living donor, an exhaustive check-up is performed, and the detailed gynecological history is known, but, in this case, the potential risk of complications for the donor is a major limitation [30]. The use of a uterus from a deceased donor has the main advantage of not causing complications for the donor, but the surgery is not scheduled and has to be performed in emergency settings, not optimal conditions. Furthermore, not all recipients have a potential compatible living donor, and, in France, a living donor must come from a close family circle. In contrast, in the USA, living organ donation is a voluntary process and not restricted to the immediate family of the recipient, resulting in more potential living donors being available. In terms of the success of the transplant (i.e., return of menstruation), based on the published data, using a graft from a living donor or from a deceased donor yields the same result, respectively, 82% (38/46) and 80% (8/10) [31].

Using deceased donors is a more recent method, and more perspective is needed. Therefore, in order to offer uterine transplantation to as many potential recipients as possible, both strategies should be developed.

Our study is not comparable to the US practices, because regulations in France are different. In the USA, the donor’s consent is explicit, contrary to France, where consent is presumed. This difference can increase the potential number of donors. In France, the uterus is not considered to be an organ but a biological tissue. In practice, this difference requires specific authorization to take the uterus in a multi-organ procurement. This requirement for additional authorization could further reduce the number of grafts available. Despite these differences, the potential number of donors is very limited and cannot meet the demand of all patients requesting a uterine transplant.

In conclusion, the feasibility of uterus transplant using a deceased donor has been proven, but the potential number of grafts is very limited. In France, only 21 are available per year, i.e., 2.8 PMP. To increase the potential number of grafts, expanding the donor criteria is necessary but should be implemented carefully, as it increases the risk of poor-quality grafts or perioperative complications (graft thrombosis for example). A meticulous pretransplant check-up must be performed on the donor, and gynecological history should be sought where possible. The restricted number of grafts requires a complementary strategy using living and deceased donors, but the constraints of each approach are not the same and must be considered before setting up a uterine transplant program. A deceased donor program can be developed in order to meet the demand of all women with absolute uterine infertility.

## Figures and Tables

**Figure 1 jcm-11-00730-f001:**
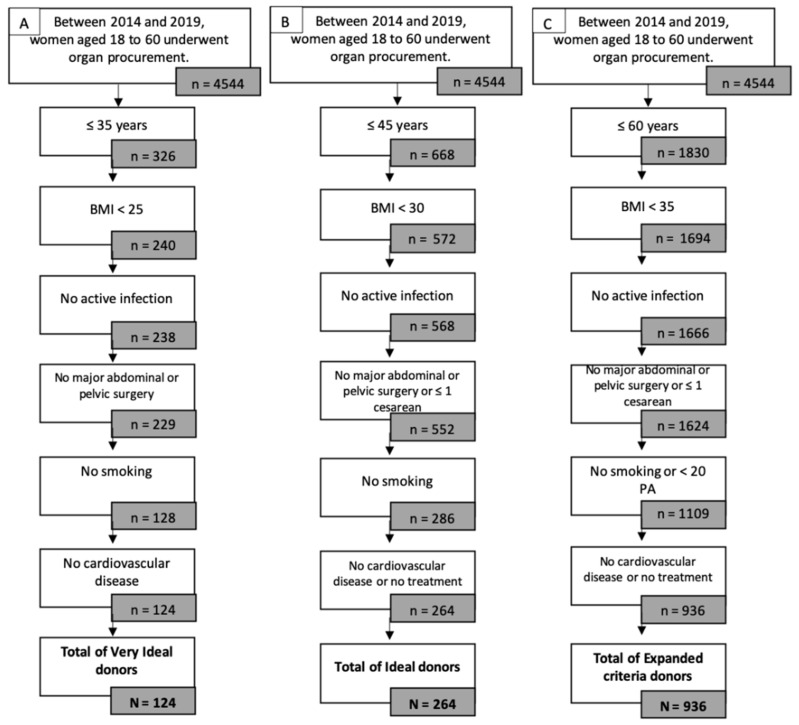
Potential donors by eligibility criteria: age, BMI, viral status, cardio-vascular disease and smoking status. (**A**) Very Ideal donor; (**B**) Ideal donor; (**C**) Expanded Criteria Donor.

**Table 1 jcm-11-00730-t001:** Advantages and disadvantages of living and deceased donors in UT.

	Living Donor	Deceased Donor
**Donor**		
Surgical risk	Ureteral injury [12]	No
Surgery duration	380 min [3]	80 min [10]
**Uterus**		
Age	Older (mother of recipient)	Potentially younger
Morphology	Best assessment	Less comprehensive assessment
Infection	Complete assessment	No complete assessment
Vascular pedicle length	Smaller	Longer: internal iliac patch
Vascular pedicle injury	Higher	Lower but still exists
**Recipient**		
Risk of rejection	Lower	Higher
Histocompatibility	Best (related donor)	Less good

**Table 2 jcm-11-00730-t002:** Inclusion criteria for very ideal, ideal, and expanded criteria deceased donors, as well as exclusion criteria for uterus transplantation.

Inclusion Criteria for Very Ideal, Ideal, and Expanded Criteria Deceased Donors			
	Very Ideal Donor	Ideal Donor	Expanded-Criteria Donor
**Age (y)**	18–35	18–45	18–60
**BMI (kg/m^2^)**	<25	<30	<35
**Surgery**	No major abdominal or pelvic surgery, No cesarean section	No major abdominal or pelvic surgery including more than one cesarean section	
**Infection**	No active infection, no seropositive for HIV, HBV, HCV, or syphilis		
**Smoking**	No		<20 PY
**Cardiovascular diseases**	No	No, or no treatment	
**Exclusion Criteria for Uterus Transplantation**			
**Age (y)**	≤18 or ≥60		
**BMI (kg/m^2^)**	≥35		
**Surgery**	Major abdominal or pelvic surgery		
**Infection**	Active infection or active HSV, seropositive for HIV, HBV, HCV, syphilis		
**Smoking**	>20 PY		
**Cardiovascular diseases**	severe arterial hypertension, Diabetes mellitus type I and II by medical history.		

**Table 3 jcm-11-00730-t003:** Number of potential very ideal donors, ideal donors, and expanded criteria donors.

	Very Ideal Donor	Ideal Donor	Expanded Criteria Donor
**2014** **number**	21	53	151
**PMP**	2.9	4.6	8.2
**2015** **number**	20	45	153
**PMP**	2.8	3.9	8.3
**2016** **number**	21	43	181
**PMP**	2.9	3.7	9.9
**2017** **number**	20	38	143
**PMP**	2.8	3.3	7.9
**2018** **number**	27	48	168
**PMP**	3.8	4.2	9.3
**2019** **number**	15	37	140
**PMP**	2.1	3.3	7.7
**Per year during** **2014–2019**	21	44	156
**PMP**	2.8	3.8	8.6

**Table 4 jcm-11-00730-t004:** Inclusion criteria of deceased donor according to research teams.

	Age (Years)	Viral Wtatus (HIV/HCV/HBV)	BMI kg/m^2^	Smoking Wtatus	Cardio-Vascular Disease	Cancer	Gynecological History	Morphology of Uterus
Cleveland [26]	Pre-menopausal	negative	<30	No	NO	UK	No history of infertility, multiparous donor preferred	Normal (US/CT/MRI)
Dallas [27]	<35	negative	<30	No		0	One full-term live birth	Normal US CT
Czech Republic [11]	18–60	NC	NC	NC	0	0	Maximum of four deliveries, including one cesarean section	US, HSC

US: ultrasonography, CT: computed tomography, MRI: magnetic resonance imaging, HSC: Hysteroscopy, UK: unknown.

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
