# Peer review of "Procurement of Uterus in a Deceased Donor Multi-Organ Donation National Program in France: A Scarce Resource for Uterus Transplantation?"

_jcm, 2022, doi:10.3390/jcm11030730_

Round 1

Reviewer 1 Report

In summary, this is a very well-written article with a relatively complete "Literature Review" section. However, own data are only available to a limited extent.

In my opinion, inclusion criteria are vague (why e.g. DM type 1 excluded, or only in the case of end organ manifestations?) and too few parameters are recorded. 

A standardized check of the donor before removal and expansion of the inclusion and exclusion criteria developed within the framework of the international UTX network and the data of the UTX registry would be recommended. Perhaps the authors would like to make suggestions here?

What database was used to determine the suitability of the donors (very ideal, ideal, extended criteria)?

It would be helpful to compare the pros and cons of the two types of donors in a table

It would be helpful to compare the advantages and disadvantages of the two types of donors in a table and to emphasize the limits and strengths of the work more clearly

"In order to increase the number of available uteri from deceased donors, the inclusion criteria for donors should be expanded” (page 6 of 9). - This statement should be justified in more detail with data and the advantages and disadvantages presented.

To increase the number of available uteri from deceased donors, the inclusion criteria for donors should be expanded "(page 6 of 9). This statement should be justified in more detail with data and the pros and cons should be presented.

The parameters, numerical values and definitions shown in the text, in tables and figures should be checked for consistency and agreement.

The authors are advised to use the (Microsoft Word) template file for recommended formatting requirements.

Abbreviations and Acronyms should be defined the first time they appear and added in parentheses after the written-out form; they should be kept constant throughout the manuscript.

For better readability and clarity, consideration should be given to place the reference numbers in square brackets [ ], and before the punctuation.

The authors are advised to have the manuscript read by a native English-speaking colleague or an English editing service.

Reviewer 2 Report

In this paper, Dion and colleges analyzed the data in the national database, then proved that the potential number of the uterus grafts from deceased donor is very limited in France. This paper would be significantly strengthened by collecting and analyzing large number of the data from CRISTAL, the national data base in France.

However, it is well known that the numbers of the organs for the transplants are insufficient from deceased donor all over the world, including uterus transplant. The result of this article, the analyzed data showing the shortage of uterus graft from deceased donor in the France, will not have any great impact to move the uterine transplant field forward. And some hypothesis, definitions are not well written, thus those would be recommended to well revised.

I have outlined additional opportunities to provide clarity and improve the manuscript below.

Specific comments

  1. “Harvest” is one of the unacceptable words in the field of transplant. Please consider to change them to “Procure” or the other preferred terms. For further guidance, refer to the glossary maintained by the Organ Procurement and Transplantation Network at https://optn.transplant.hrsa.gov/resources/glossary.
  2. 2.1. Survey development: You may not need approval from ethics’ board, but it would be better to write the approval number and/or the date of approval from the French National Agency of Biomedicine for these data collection. In addition, it might be better to write the exact duration of the data, not only year, but also month and date.
  3. 2.2. Inclusion criteria: In the reference #20, they classified in two categories for incursion criteria: standard criteria donor and expanded criteria donor. It looks like standard criteria donor is divided in two categories on this paper: Very Ideal donor and Ideal donor. As these categories are new in the transplant field, please explain the reason in detail if these new categories are needed, including the benefit, scientifically and statistically if possible. Otherwise please consider to use the incursion criteria in the same way as reference #20, standard criteria donor and expanded criteria donor, and then change the Table 1, Table 2, and Figure 1.
  4. Tobacco: It is understandable that the tobacco is the risk of stroke. However, it is not described the reason <20PY smokers to be the expanded criteria donor. Please describe the reason 20PY is the exact number for the ECD. “ECD: ≤5 cigarettes/d” is written on reference #20. Moreover, it is written on the conclusion of the reference #21 “Across the full range of cigarettes/day, smoking fewer cigarettes/day for longer durations was more deleterious than smoking more cigarettes/day for shorter durations.”
  5. Discussion: The success of the uterus transplant means many things, (e.g., surgical success, long-term graft survival, pregnancy, delivery, hysterectomy after delivery). It would be better to explain the definition “In terms of success of the transplant”, if possible.
  6. Others: It would be new and helpful to know the comparison of some data between uterus and the other organs, (e.g. waiting time, numbers of the waiting list, sufficiency rate, PMP of each organ donors, inclusion(standard, expanded)/exclusion criteria, etc.) if you describe them on this paper.

Thank you for the opportunity to review.

Author Response

In this paper, Dion and colleges analyzed the data in the national database, then proved that the potential number of the uterus grafts from deceased donor is very limited in France. This paper would be significantly strengthened by collecting and analyzing large number of the data from CRISTAL, the national data base in France.

However, it is well known that the numbers of the organs for the transplants are insufficient from deceased donor all over the world, including uterus transplant. The result of this article, the analyzed data showing the shortage of uterus graft from deceased donor in the France, will not have any great impact to move the uterine transplant field forward. And some hypothesis, definitions are not well written, thus those would be recommended to well revised.

Response :

The authors wish to thank the reviewer for the quality of the comments and suggestions made. The authors therefore clarified the definitions and clumsiness of the terms used. we have taken into account all of the following comments and modified the manuscript accordingly.

I have outlined additional opportunities to provide clarity and improve the manuscript below.

Specific comments

  1. “Harvest” is one of the unacceptable words in the field of transplant. Please consider to change them to “Procure” or the other preferred terms. For further guidance, refer to the glossary maintained by the Organ Procurement and Transplantation Network at https://optn.transplant.hrsa.gov/resources/glossary.

Response : The authors agree with this remark and change the word “Harvest” by “Procure” in all the text.

  1. 2.1. Survey development: You may not need approval from ethics’ board, but it would be better to write the approval number and/or the date of approval from the French National Agency of Biomedicine for these data collection. In addition, it might be better to write the exact duration of the data, not only year, but also month and date.

Response  :The authors agree with this comment and they specified the dates during which the study was carried out and the date of approval from the French National Agency of Biomedicine in the paragraph 2.1. Survey development.

We analyzed the data reported in the CRISTAL database regarding all women aged 18 years and older on whom organ harvesting procurement has been performed between January 1, 2014 and December 31, 2019. This study is exempt from approval an ethics’ board and was approved on March, 9 2020 from the French National Agency of Biomedicine.

  1. 2.2. Inclusion criteria: In the reference #20, they classified in two categories for incursion criteria: standard criteria donor and expanded criteria donor. It looks like standard criteria donor is divided in two categories on this paper: Very Ideal donor and Ideal donor. As these categories are new in the transplant field, please explain the reason in detail if these new categories are needed, including the benefit, scientifically and statistically if possible. Otherwise please consider to use the incursion criteria in the same way as reference #20, standard criteria donor and expanded criteria donor, and then change the Table 1, Table 2, and Figure 1.

Response : The authors agree with this remark, UTx is new new surgical approach in the transplantation community.

There is no consensus inclusion criterion for brain-dead donors. We have taken up the inclusion criteria published by Kristek et al., the very ideal donor patient category has been added in order to know the number of patients meeting strict inclusion criteria in order to obtain better quality grafts. The very ideal donor patient category is different from the ideal donor patient only by 2 criteria: the BMI < 25 and the absence of antecedent caesarean section. 
This category of patient is given for information and may be deleted if it disturbs the reading of the article.

The authors have added the following explanation in the material and method part : Compare to the Kristek et al. study, we arbitrarily added a patient category: the vary ideal donor. This addition was made for the sole purpose of knowing the number of grafts available by having very strict donor inclusion criteria with a BMI < 25 and the absence of antecedent caesarean section.

In this paper Kristek et al. mentioned that the criteria could be modified with the technical evolution : “Certainly “working” criteria will be useful that may be subject to change as more experience is gained. “

  1. Tobacco: It is understandable that the tobacco is the risk of stroke. However, it is not described the reason <20PY smokers to be the expanded criteria donor. Please describe the reason 20PY is the exact number for the ECD. “ECD: ≤5 cigarettes/d” is written on reference #20. Moreover, it is written on the conclusion of the reference #21 “Across the full range of cigarettes/day, smoking fewer cigarettes/day for longer durations was more deleterious than smoking more cigarettes/day for shorter durations.”

Response : The authors agree with this comment. In the Cristal database, tobacco consumption is given in packet-year and not in cigarettes/day. To adapt, the authors sought a cut-off of tobacco consumption that could lead to an overrisk of cardiovascular event. Lubin et al. described a greater increase in cardiovascular pathology starting from 20PY, which is why this baked-off was defined.

The authors added this sentence in the material and method section: In the Cristal database, tobacco consumption is given in packet-year and not in cigarettes/day. To adapt, the authors sought a cut-off of tobacco consumption that could lead to an overrisk of cardiovascular event. Lubin et al. described a greater increase in cardiovascular pathology starting from 20PY, which is why this baked-off was defined.

  1. Discussion: The success of the uterus transplant means many things, (e.g., surgical success, long-term graft survival, pregnancy, delivery, hysterectomy after delivery). It would be better to explain the definition “In terms of success of the transplant”, if possible.

Response : The authors agree with this remark, the success of the uterus transplantation is defined by the return of menstruation.

The authors added this definition in the discussion  : In terms of success of the transplant (i.e. return of menstruation), based on the published data, using a graft from a living donor or from a deceased donor yields the same result, respectively 82% (38/46) and 80% (8/10)[27].

  1. Others: It would be new and helpful to know the comparison of some data between uterus and the other organs, (e.g. waiting time, numbers of the waiting list, sufficiency rate, PMP of each organ donors, inclusion (standard, expanded)/exclusion criteria, etc.) if you describe them on this paper.

Response : The authors agree on that but they did not find similar data for the analysis of another organ.

ECD were initially defined for kidney transplantation and are very specific to it. These criteria are therefore not applicable to the UT.

Here are the criteria from Organ Procurement and Transplantation Network :

“A kidney donated for transplantation from any brain dead donor over the age of 60 years; or from a donor over the age of 50 years with two of the following: a history of hypertension, the most recent serum creatinine greater than or equal to 1.5 mg/dl, or death resulting from a cerebral vascular accident (stroke). This definition applies to the allocation of deceased donor kidneys.

Round 2

Reviewer 1 Report

Reviewer comment on January 9, 2022

I appreciate the revisions that the authors have added in such a short time. However, with respect to the importance of the topic and the journal, also the minor issues noted below should be fixed.

Best regards.

I.Inconsistencies

I.a. Definitions and cut-offs: Age

  • Abstract: „We analyzed the data reported in the CRISTAL database regarding all women aged 18 years and older …“
  • Material and Methods (2.1) „…We analyzed the data reported in the CRISTAL database regarding all women aged 18 years and older …“
  • Table 2: Inclusion criteria:
    • „Age 18-35“
    • „Age 18-45“
    • „Age 18-60“
  • Design of the study (2.4):
    • „…women aged 18 to 60.“
    • „…by age category is represented in figure 1.“
  • Figure 1:
    • „≤ 35 years“
    • „≤ 45 years“
    • „≤ 60 years“

I-b. Inconsistency of cut-offs for age

- in table 2 and figure 1 with the definitions in Abstract / Material and Methods / Table 2 (Inclusion criteria)

  • Table 2: Exclusion criteria: Age „≤ 18 or ≥ 60“
  • Figure 1:
    • „…women older than 18 years…“
  • Page 5 of 11:
    • „…women under 35…“
    • „…women under 45…“
    • „…women under 60…“

II.Typing errors

  • Figure 1: „underwent organ harvesting“ [procurement]
  • Abstract: „Keys words“ [Key words]
  • Abstract: „Poeple“ [People]
  • Abstract: „extend criteria“ [Extended Criteria Donor /ECD]
  • Key words: „extend criteria“ [Extended Criteria Donor /ECD]
  • Page 6 of 11: „extend criteria“ [Extended Criteria Donor /ECD]
  • Discussion: „extend criteria“ [Extended Criteria Donor /ECD]
  • Page 7 of 11: „extend criteria“ [Extended Criteria Donor /ECD]
  • Table 1: „…Living and Deceased Donor …“ [Donors]
  • Table 2, Exclusion criteria: „Smoke“ [Smoking]
  • Table 4, Smoking status: „No drug“ [no]
  • Table 4: „NC“ [explain this term]
  • Figure 1: „Smoke“ [Smoking]
  • Figure 1: „…no treat“ [no treatment]
  • Page 1 of 11: „…contribute equally in present work…“ [contributed equally to this work]
  • Page 2 of 11: „…could be candidate…“ [candidates]
  • Page 3 of 11: „Materiel …“ [Material]
  • Page 7 of 11:
    • „…for all recipient …“ [recipients]
    • „…regarding to UT…“ [UTx]
    • „…for extend criteria…“ [extended]
    • „…deceased donors the in Dallas team.“ […deceased donors in the Dallas team.]
  • Page of 11:
    • „…UT all over the world…“ [UTx all over the world]
    • „…in UT field…“ [in UTx field]
  • Material and methods (2.1): „…March, 9 2020 …“ [March 9, 2020]
  • Material and methods (2.2):
    • „…Compare to the…“ [Compared]
    • „…vary ideal donor…“ [very]
    • „…the Cristal database…“ [CRISTAL]
    • „…packed year…“ [pack years]
    • „…overrisk of cardiovascular event…“ [higher risks of cardiovascular events]
    • „…baked off…“ [cut-off]
    • „…95% IC…“ [95% CI]
  • Acknowledgments: „biomedicine agency“ [Biomedicine Agency]
  • Disclosure: „.no conflicts of interest.“ […no conflict of interest]
  • Abbreviations: „UT“ [UTx]
  • Abbreviations: „Per Million Poeple“ [Per Million People]

Reviewer 2 Report

Thank the authors for the revises. Your revised article seems so much better than the first one. You do not need to delete the category of very ideal donor, because you added appropriate explanation.

However, you have not so clearly answered about the last question (Others: the comparison of some data between uterus and the other organs). You might find these data from written below.

1) From OPTN data (U.S. data):

https://optn.transplant.hrsa.gov/data/view-data-reports/national-data/

2) From OPTN/SRTR Annual Data Report (2019 U.S. data):

https://onlinelibrary.wiley.com/toc/16006143/2021/21/S2

= ( https://srtr.transplant.hrsa.gov/annual_reports/2019_ADR_Preview.aspx )

3) PMP of each organ donors (2019 France data):

https://www.irodat.org/?p=database&c=FR&year=2019#data

You can find the other years’ data in France as well.

Too much data might make confusions to the readers. Thus, as the number of PMP of expected uterus donors is the one of the key points of this article, it would be recommended to assess them comparing with the other organs in France using above 3) data, at the least. Additionally, it would be better to describe your plan after the assessment in Discussions, if possible. (e.g. (After showing the numbers of PMP of the other organ donors data (with reference #)), the number of PMP of expected uterus donors is smaller like pancreas donors, compared with the other organs. One of the reasons of the small number of the pancreas donors could be … (e.g. high graft loss ratio 10-20% due to thrombosis (with reference #), or/and etc.) like uterus transplant. However, if the graft loss ratio become much better than pancreas data (or the current uterus data) in the near future, … (e.g. these criteria may be changed to increase the number of DD uterus transplants /or these criteria should be kept to prevent graft loss anyway, etc.)).

Regarding pancreas transplant data; e.g. https://doi.org/10.1148/radiol.2015150437

“Vascular thrombosis is the most common nonimmunologic cause of early pancreas transplant failure, and it usually occurs within the splenic vein, with a reported loss rate of 5%–29% (2–9).”

Please accept my sincere apology, if you feel discomfort to my too much suggestions.

Thank you for the opportunity to review again.

The last comments: You have done so great works, so I hope this paper will be used for the reference in the field of DD uterus transplant whole overt the world, and then you will be expected to report the French DD uterus transplant data in the near future. Good luck to you.
